# Generalised analytical method unravels framework-dependent kinetics of adsorption-induced structural transition in flexible metal−organic frameworks

Yuta Sakanaka[1], Shotaro Hiraide [1]✉, Iori Sugawara[1], Hajime Uematsu[1], Shogo Kawaguchi [2], Minoru T. Miyahara [1] & Satoshi Watanabe [1]✉

Flexible metal−organic frameworks (MOFs) exhibiting adsorption-induced structural transition can revolutionise adsorption separation processes, including $CO_2$ separation, which has become increasingly important in recent years. However, the kinetics of this structural transition remains poorly understood despite being crucial to process design. Here, the $CO_2$-induced gate opening of ELM-11 ($[Cu(BF_4)_2(4,4'-bipyridine)_2]_n$) is investigated by time-resolved in situ X-ray powder diffraction, and a theoretical kinetic model of this process is developed to gain atomistic insight into the transition dynamics. The thus-developed model consists of the differential pressure from the gate opening (indicating the ease of structural transition) and reaction model terms (indicating the transition propagation within the crystal). The reaction model of ELM-11 is an autocatalytic reaction with two pathways for $CO_2$ penetration of the framework. Moreover, gas adsorption analyses of two other flexible MOFs with different flexibilities indicate that the kinetics of the adsorption-induced structural transition is highly dependent on framework structure.

The separation of $CO_2$ is an urgent issue requiring immediate resolution. Amine absorption is a well-established technology for $CO_2$ separation[1–3], although its sustainable development is impeded by corrosiveness, degradation, harmful by-product emission and a large heat energy requirement[4,5]. This has prompted extensive research on low-energy separation technologies using porous materials[6]. Moreover, metal−organic frameworks (MOFs), which can be tuned by appropriate selection of a metal ion and organic linker, have been widely studied[7,8].

Flexible MOFs undergo structural transition in response to guest molecule adsorption, via a unique 'gate opening' or 'breathing' mechanism[9–11]. This process exhibits high selectivity because of their flexibility and high working capacity induced by an abrupt increase in adsorption at the gate-opening pressure ($P_{gate}$). ELM-11 ($[Cu(BF_4)_2(bpy)_2]_n$; bpy = 4,4'-bipyridine)[12,13], which demonstrates the typical gate-opening behaviour, can significantly improve $CO_2/CH_4$ separation via pressure swing adsorption[14], and its application in adsorption-based separation processes is currently being experimentally and/or theoretically investigated[15].

Elucidating the structural transition kinetics of flexible MOFs is essential for performing process simulations. However, only a few studies have been published on this topic thus far[16,17]. Theoretical models, such as the pore diffusion model and its first-order approximation (i.e. linear driving force model), have been extensively used to describe the adsorption kinetics of conventional adsorbents[18–20]. However, such models cannot be applied to flexible

[1]Department of Chemical Engineering, Kyoto University, Nishikyo, Kyoto 615-8510, Japan. [2]Japan Synchrotron Radiation Research Institute (JASRI), SPring-8, 1-1-1 Kouto, Sayo, Hyogo 679-5198, Japan. ✉e-mail: hiraide@cheme.kyoto-u.ac.jp; nabe@cheme.kyoto-u.ac.jp

MOFs because gate opening involves a phase transition from a non-porous structure. Therefore, it is vital to develop a kinetic model applicable to flexible MOFs. This model should be based on an atomistic understanding of the structural transition to ensure a one-to-one correspondence between the transition kinetics and mechanism (similar to that between the chemical reaction rate and reaction mechanism).

Recent studies have directly visualised the phenomena occurring at the nanoscale using transmission and scanning electron microscopies[21–23]. However, observing the local structural changes over time, to derive a kinetic model for adsorption-induced structural transitions from an atomic-level understanding, remains challenging. The constant-volume method[24], which involves the introduction of gases at a specific pressure and an analysis of the pressure change over time, is widely used for adsorption kinetics analyses; some studies have applied this method to flexible MOFs[25–27]. However, pressure plays a dual role in this method (as a condition variable that determines the saturated adsorbed amount and a measurement variable from which the current adsorbed amount is calculated); thus, it is challenging to use this technique for an accurate analysis.

In situ X-ray powder diffraction (XRD) analysis is widely used to study the structural transition of flexible MOFs[28–30], and time-resolved measurements enable an observation of this dynamic process[14,31–36]. Time-resolved in situ XRD (TRXRD) was used to investigate the kinetics of $CO_2$ gate adsorption on ELM-11[14]. Subsequent model fitting via trial-and-error suggested that this process follows the Kolmogorov–Johnson–Mehl–Avrami (KJMA) equation[37,38] and that the differential pressure relative to $P_{gate}$ acts as the driving force for the

transition. However, this model is empirical and lacks physical significance, i.e. it does not reflect the structural transition dynamics.

In this study, a kinetic model of structural transition was derived from TRXRD data to elucidate the structural transition mechanism. TRXRD data for the $CO_2$ gate opening on ELM-11 was obtained by increasing the gas pressure at a constant rate. The transition process could be described by an autocatalytic reaction model, which was consistent with the atomistic details of the framework structure of ELM-11. Moreover, the same analysis was conducted for $CO_2$ breathing in MIL-53(Al) ([Al(OH)(bdc)]$_n$; bdc = 1,4-benzendicarboxylate)[39] and $CO_2$ gate opening in CuFB ([Cu(fumarate)(trans-bis(4-pyridyl) ethylene)$_{0.5}$]$_n$)[40,41]. The results confirmed that the adsorption kinetics of flexible MOFs is highly dependent on their framework structure.

## Results
### TRXRD measurements

TRXRD was used to investigate the structural transition of ELM-11 upon the introduction of $CO_2$ at a constant rate. Figure 1 shows the TRXRD pattern at 0.8 kPa s$^{-1}$ and 248 K as a typical example. Initially (at 0 kPa), peaks representing the non-porous structure (closed phase) appeared. $CO_2$ was introduced after 5 s, and peaks corresponding to a $CO_2$-encapsulating structure (open phase) started to appear at ~19 s. Upon increasing the pressure, the peaks representing the closed phase gradually weakened, while those representing the open phase intensified; only the latter were observed in the XRD pattern recorded after 31 s. The time elapsed between the open-phase appearance and closed-phase disappearance was ~12 s. All the peaks in the XRD patterns recorded during the structural transition could be attributed to the

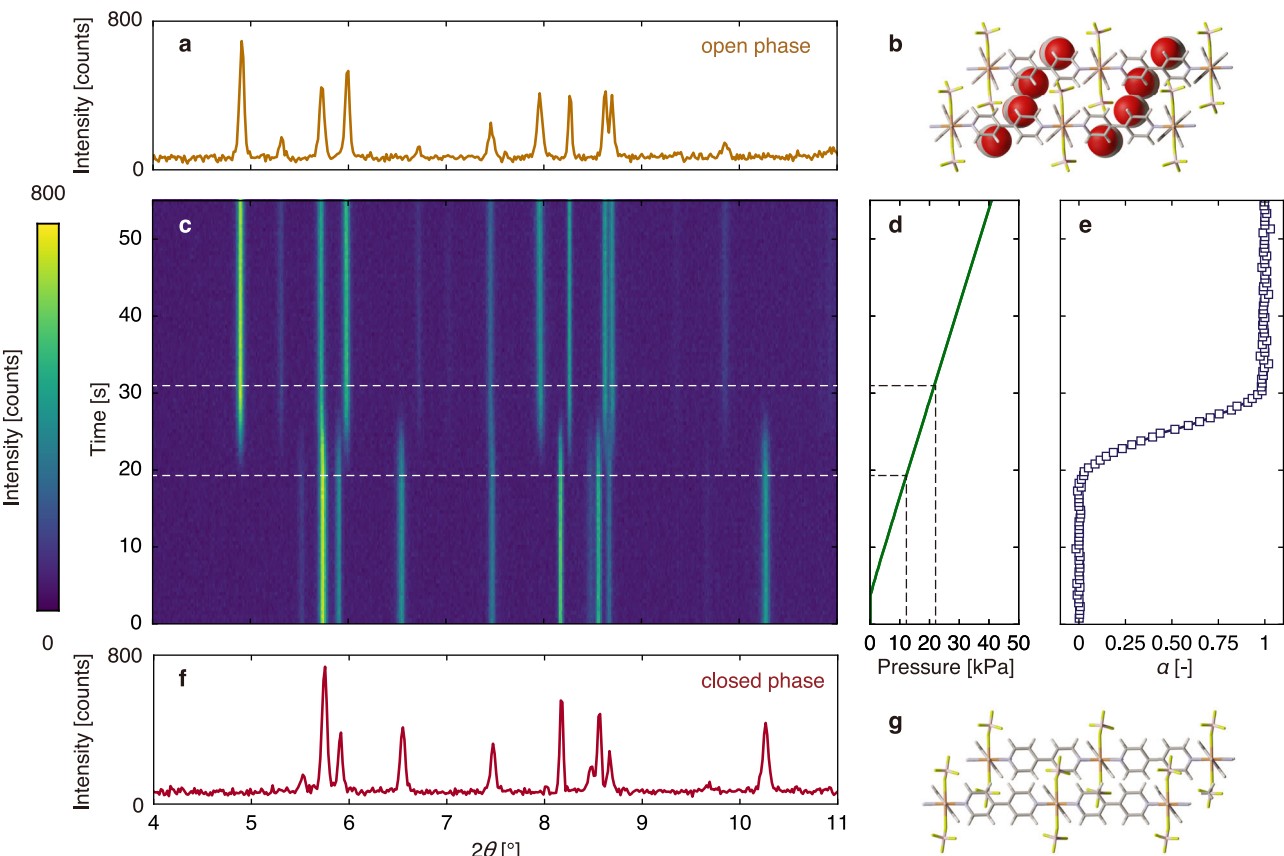

**Fig. 1 | Time-resolved in situ X-ray powder diffraction (TRXRD) data. a** XRD pattern at 42 kPa and (**b**) crystal structure of ELM-11 in the open phase. **c** Colormap of TRXRD patterns of ELM-11 at 248 K and 0.8 kPa s$^{-1}$. **d** Time evolution of the $CO_2$ pressure. $CO_2$ was introduced 5 s into the measurement at a constant flow rate of up to 42 kPa. **e** Time evolution of $\alpha$, calculated from the ratio of the XRD patterns in **a** and **f** at regular intervals of time. **f** XRD pattern at 0 kPa and (**g**) crystal structure of ELM-11 in the closed phase. In the crystal structure, the atoms are colour-coded as follows: H (white), B (pink), C (grey), N (purple), O (red), F (green) and Cu (orange).

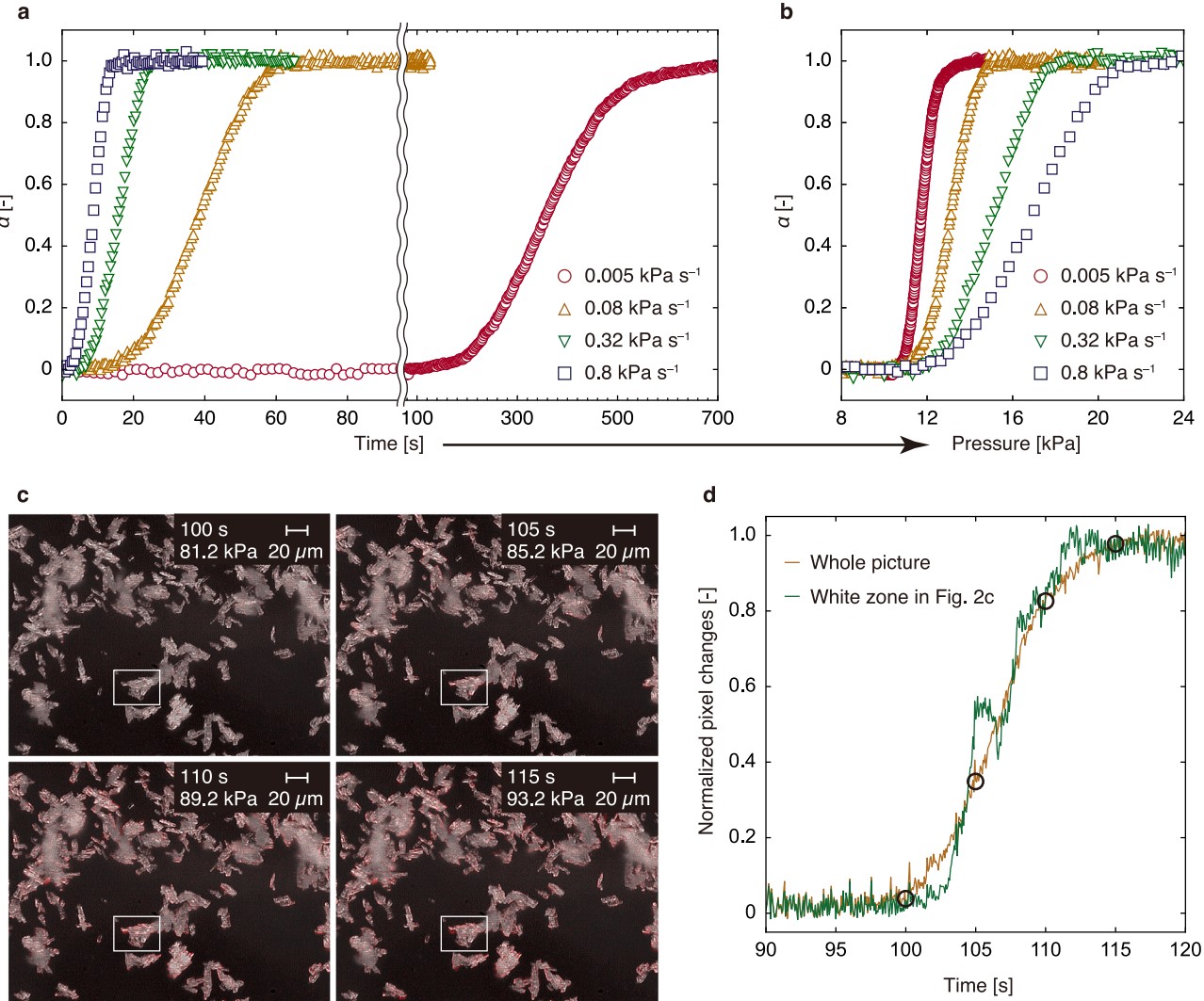

**Fig. 2 | CO$_2$ gate adsorption on ELM-11 by two types of in situ measurements.**
**a** Time evolution of $\alpha$ at 248 K and 0.005, 0.08, 0.32 and 0.8 kPa s$^{-1}$. The origin of the x-axis was set as the time at which the gas pressure was 10 kPa. The time scale was changed and a boundary was set at ~90 s because adsorption at 0.005 kPa s$^{-1}$ took a very long time. **b** Pressure dependence of $\alpha$ in **a**. For all pressurisation rates, the structural transition began when the gate opening pressure at 248 K (10 kPa) was exceeded. **c** Optical microscopy snapshots of ELM-11 particles during CO$_2$

introduction within a time range of 100–115 s at ~297 K. In each image, the pixels that have changed from the first frame (0 s) are coloured red. The corresponding movies (with and without colouring) are shown in Supplementary Videos 1 and 2. **d** The time evolution of the normalised number of pixels altered from the initial frame (red pixels in **c**); the yellow line was analysed using the whole picture, while the green line was analysed using the region delineated by white lines in **c**. Open circles correspond to the points shown in **c**.

closed or open phase; thus, there was no intermediate state with long-range periodicity.

Because of the absence of an intermediate state, the kinetics of structural transition was analysed using the fraction of the open phase, i.e. fraction transformed ($\alpha$). Figure 1e shows a sigmoidal $\alpha$ profile, which was obtained by analysing the data shown in Fig. 1c (the reproducibility of measurement is discussed in Supplementary Note 1). Figure 2a shows the time evolution of $\alpha$ with the pressure increasing at different rates (0.005, 0.08, 0.32 and 0.8 kPa s$^{-1}$). The completion time of the structural transition increased as the pressurisation rate decreased. Subsequently, the profiles were re-plotted against pressure by multiplying the data with the pressurisation rate (Fig. 2b). Under all conditions, the transition commenced at ~10 kPa, which was close to the $P_{gate}$ value at 248 K (the adsorption pressure at half-maximum uptake[42] is 10.3 kPa; see Supplementary Fig. 1). Thus, ELM-11 does not transform at pressures lower than $P_{gate}$. However, the pressure at the end of the transition varied with the pressurisation rate (14 and 22 kPa at 0.005 and 0.8 kPa s$^{-1}$, respectively), indicating that at pressures slightly greater than $P_{gate}$ (red circles in Fig. 2b), the structural

transition was slow and required a long time for completion (red circles in Fig. 2a).

Because XRD reflects the average collective behaviour of particles, it is impossible to distinguish between the transition of a single particle or variations in the transition time of each particle. In fact, Miura et al. reported a significant discrepancy between the results obtained by TRXRD and microscopic observations in the case of instantaneous vapour dosing of the DUT-8 series[43]. To address this issue, TRXRD was conducted by varying the inner diameter of a glass capillary filled with ELM-11. The capillary diameter and consequently, gas diffusion through the packed bed, did not influence the structural transition (Supplementary Fig. 2). Figure 2c and Supplementary Videos 1 and 2 show the volume expansion of ELM-11 particles because of gate opening when the CO$_2$ pressure was increased at 0.8 kPa s$^{-1}$ and room temperature (~297 K). Volume expansion required ~15 s, which was almost the same as the transition time recorded using TRXRD. More specifically, Fig. 2d shows the time evolution of the number of pixels altered from the initial frame (highlighted as red pixels in Fig. 2c), revealing a sigmoidal curve

resembling the TRXRD results (Fig. 2a). This trend remains consistent when the same analysis is applied to the region delineated by white lines in Fig. 2c (represented by the green line in Fig. 2d), indicating that the time evolution indicated by TRXRD reflects single-particle structural transition kinetics. The discrepancy between the current findings and those reported by Miura et al.[43] could be explained by the gradual increase in pressure during our measurements compared to their instantaneous pressure increase, or by the use of different types of flexible MOFs.

## Development of a transition kinetic model

The collected data were used to analyse the structural transition rate of ELM-11. Analogous to the analysis of combustion reactions (in which the change in weight is measured while increasing the temperature at various constant rates[44,45]), $\alpha$ was recorded while increasing the pressure at various constant rates. Therefore, the time evolution of $\alpha$ is

$$\frac{d\alpha}{dt} = f(\alpha, P), \tag{1}$$

where $f$ is a function in which the dynamic mechanism of structural transition is inherent. The rate of pressure increase is

$$\frac{dP}{dt} = v_p, \tag{2}$$

where $v_p$ is the constant pressurisation rate (0.005, 0.08, 0.32 and 0.8 kPa s$^{-1}$). Dividing Eq. (1) by Eq. (2) yields the following expression indicating the evolution of $\alpha$ with pressure:

$$f(\alpha, P) = v_p \frac{d\alpha}{dP}, \tag{3}$$

where $d\alpha/dP$ can be determined from the measurements (Fig. 3a). Since $\alpha$ and $P$ are variables with a one-to-one correspondence, $f(\alpha, P)$ can be evaluated using Eq. (3) (Fig. 3b).

Figure 3c shows the variation of $f(\alpha, P)$ with pressure, with fixed $\alpha$ values in the range of 0.1–0.9. The plot is linear and intersects the $x$-axis near $P_{gate}$, which explains the experimentally observed slow structural transition near $P_{gate}$. Because the slope and $x$-intercept are functions of $\alpha$ (hereinafter designated as $g(\alpha)$ and $h(\alpha)$, respectively), $f(\alpha, P)$ can be written as

$$f(\alpha, P) = g(\alpha)(P - h(\alpha)) \tag{4}$$

Figure 3d shows that the relationship between $\alpha$ and $g(\alpha)$ is parabolic. Thus, the part of the structural transition rate represented by $g(\alpha)$ can be characterised by an autocatalytic reaction,

$$g(\alpha) = (k_1\alpha + k_2)(1 - \alpha), \tag{5}$$

where $k_1$ and $k_2$ are the rate constants. Figure 3d (solid curve) shows the good fit of the experimental data to Eq. (5), with $k_1$ and $k_2$ values of 0.075 and 0.014 kPa$^{-1}$ s$^{-1}$, respectively. According to Eq. (4), $h(\alpha)$ is the pressure at which the structural transition rate, $f(\alpha, P)$, becomes zero, and it should be equal to the $P_{gate}(\alpha)$ indicated by the adsorption isotherms. Therefore, $h(\alpha) = P_{gate}(\alpha)$ was extracted from the adsorption isotherm measurements (see Methods). Figure 3e shows the good agreement between the $P_{gate}(\alpha)$ obtained from the adsorption isotherms (solid curve) and the $x$-intercepts in Fig. 3c (markers). Consequently, the time evolution of $\alpha$ obeys the following model (Model AC) for $CO_2$ gate adsorption on ELM-11:

$$\frac{d\alpha}{dt} = (k_1\alpha + k_2)(1 - \alpha)\left(P - P_{gate}(\alpha)\right). \tag{6}$$

Figure 4 shows the resultant theoretical curves of Model AC. Although the $P_{gate}$ function was derived from different adsorption isotherm measurements, Model AC reproduces the experimental data well at different temperatures (223, 248 and 273 K). The rate constants were negligibly influenced by temperature and did not follow the typical Arrhenius-type relationship (Supplementary Note 2). Thus, the parameters of the kinetic model can be determined from a single temperature condition, which would be useful for engineering applications. Note that this result is not contradictory to our previous report, which stated that the transition rate of ELM-11 can be explained by the KJMA equation[14]. This is because the autocatalytic reaction and KJMA equation share the same concept: the $k_1\alpha$ term in the autocatalytic reaction and the term involving the power of time ($t^n$) in the KJMA equation indicate the acceleration of the reaction as time progresses. Therefore, it is known that the KJMA equation yields a curve similar to that of an autocatalytic reaction[46].

## Dynamics of structural transition

A suitable kinetic model should provide some insights into the structural transition mechanism. Therefore, the proposed model was used to describe the structural transition dynamics. First, the driving force, i.e. the differential pressure between the bulk pressure $P$ and $P_{gate}$, was considered. We concluded that this differential pressure is essentially identical to the osmotic free energy change between the closed and open phases, $\Delta\Omega^{os}$. An adsorption-induced structural transition occurs at a pressure where $\Delta\Omega^{os}$ is zero (i.e. $P_{gate}$)[47]. Using a simple approximation (see Methods), $\Delta\Omega^{os}$ can be transformed as follows:

$$\Delta\Omega^{os} \simeq \left(P - P_{gate}\right)\left(\Delta V^{host} - V_m\Delta n\right), \tag{7}$$

where $V_m$ is the molar volume of the external gas, and $\Delta V^{host}$ and $\Delta n$ are the differential volume and amount adsorbed between the open and closed phases, respectively. This indicates that $\Delta\Omega^{os}$ is the driving force of the structural transition.

While the differential pressure term determines the ease of structural transition, $g(\alpha)$ should reflect the structural transition mechanism inside the crystal, i.e. the dynamic behaviour. Here, the $g(\alpha)$ for ELM-11 followed an autocatalytic reaction model, in which the reaction product (open phase after the transition) acts as a catalyst to accelerate the reaction (closed-to-open transition). To elucidate the origin of this accelerating effect, the atomic structure of ELM-11 was investigated in detail. Figure 5a, b show side-view snapshots of ELM-11 in the closed and open phases, respectively. ELM-11 is composed of 2D square grid layers consisting of copper ions and bpy stacked via BF$_4$. The layer spacing increases by 30% (0.446–0.578 nm) upon gate opening. With the rotation of bpy and because of layer spacing flexibility, $CO_2$ molecules are expected to penetrate the space between the stacked layers. Additionally, there is another path in the stacking direction, composed of 2D square grids connected by bpy and copper ions (Fig. 5c, d). Thus, $CO_2$ molecules can penetrate ELM-11 crystals via two pathways. Path A is the $CO_2$ passage in the direction horizontal to the ELM-11 layers, in which $CO_2$ molecules spread between the layers, accompanied by bpy rotation. Path B is the $CO_2$ passage in the stacking direction, in which $CO_2$ molecules penetrate the 1D channels in the interior of the lattice. We confirmed the presence of these two pathways by conducting molecular dynamics simulations based on generic neural network potentials[48] (Fig. 5e). Figure 5f illustrates the structural transition mechanism, considering both the kinetic model and atomic structure. Penetration via Path B occurs by the layer-by-layer permeation of $CO_2$ molecules, thereby leading to a low structural transition rate. However, the transition requires only a few tens of seconds to complete, according to TRXRD, indicating Path A as the predominant pathway (Fig. 5f(i)). Notably, a $CO_2$ molecule penetrating via Path A can move to an adsorption site in the adjacent layers by passing through Path B (Fig. 5f(ii)). The movement of $CO_2$ molecules above or below a

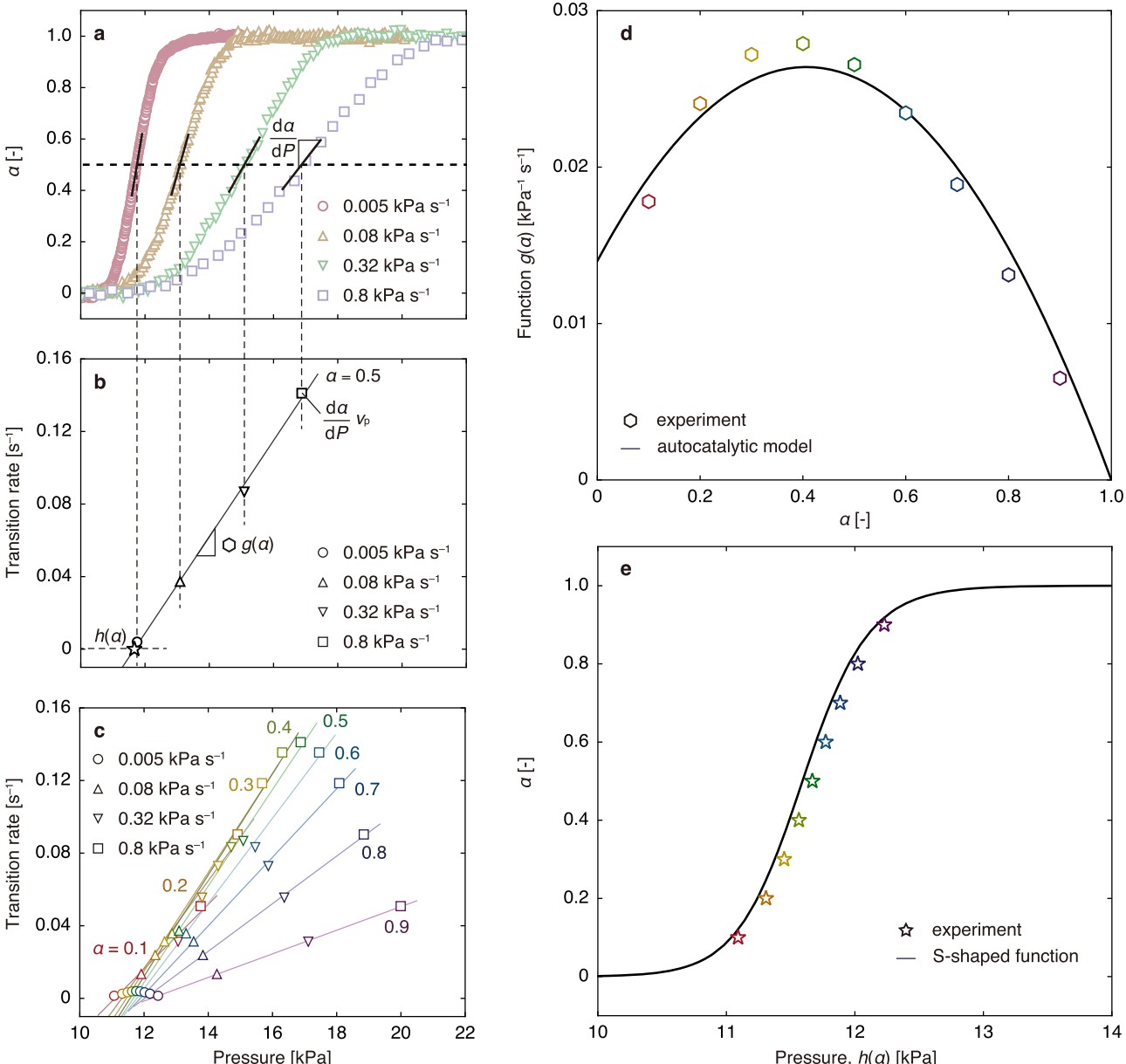

**Fig. 3 | Derivation of the structural transition rate model. a–c** Pressure dependence of the structural transition rate at various $\alpha$. **a, b** show an example of analysis for $\alpha = 0.5$, in which the derivative values of $\alpha$ with respect to pressure are extracted from the time-resolved in situ X-ray powder diffraction (TRXRD) results at 0.005, 0.08, 0.32 and 0.8 kPa s⁻¹, and the d$\alpha$/d$P$ values are obtained by multiplying the pressurisation rate. The solid lines in **b** and **c** are obtained by linear least-squares fitting, where the slope represents the function $g(\alpha)$, and the $x$-intercept represents the function $h(\alpha)$. **d** Relationship between the function $g(\alpha)$ obtained from **c** and $\alpha$. The solid curve in **d** is obtained by least-squares fitting of the autocatalytic reaction model (Eq. (5)). **e** Relationship between $\alpha$ and the function $h(\alpha)$ obtained from **c**. The solid curve is obtained from the adsorption isotherm, as shown in Supplementary Fig. 1.

layer causes a temporary semi-opening of the layer, which can promote $CO_2$ penetration via Path A of that layer (Fig. 5f(iii)). This atomistic insight is consistent with the concept of autocatalysis, which indicates that the open phase promotes the transition of the closed phase. That is, Model AC suggests that the closed-to-open transition of ELM-11 occurs gradually, layer by layer, through the coexistence of closed and open layers, which is consistent with the experimental observations obtained using in situ microscopy (see Fig. 2 and Supplementary Videos 1 and 2) and a recent theoretical study employing a simplified layer-stacked MOF model[49].

### Comparison with other flexible MOFs, MIL-53(Al) and CuFB

The autocatalytic transition behaviour of ELM-11 can be attributed to the two $CO_2$ penetration pathways in its framework structure (Fig. 6a–c). Thus, different kinetic models should be derived for flexible MOFs with different frameworks. TRXRD was used to investigate $CO_2$ adsorption on MIL-53(Al) and CuFB. MIL-53(Al) exhibits a breathing phenomenon in which the large pore (LP) phase changes to the narrow pore (NP) phase (LP → NP transition) and back to the LP phase (NP → LP transition) along with $CO_2$ adsorption. In contrast, CuFB, with a mutually interpenetrating structure, exhibits gate opening because of linker rotation. Their adsorption isotherms are shown in Supplementary Figs. 3 and 4.

Figure 6d–l shows the atomistic model, schematics of the dynamic mechanism and pressure dependence of $\alpha$ (evaluated from TRXRD measurements and calculations using the derived models for MIL-53(Al) and CuFB; the TRXRD raw data are shown in Supplementary Figs. 5–7). The LP → NP and NP → LP transitions of MIL-53(Al) and gate

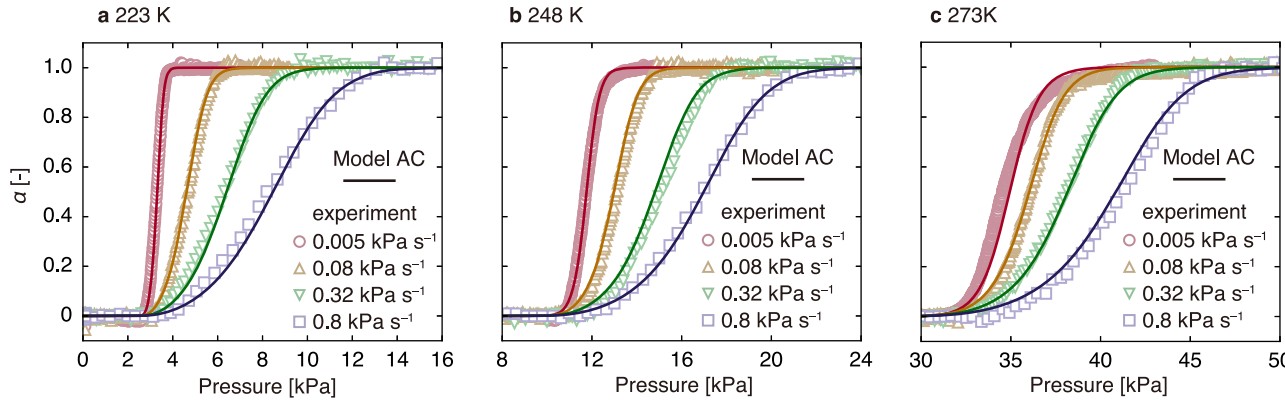

**Fig. 4 | Derived structural transition rate model and its temperature dependence. a–c** Relationships between $\alpha$ and pressure obtained from time-resolved in situ X-ray powder diffraction (TRXRD) measurements (markers) and results calculated using the structural transition rate model (curves) at 223, 248 and 273 K, respectively.

opening of CuFB did not exhibit the sigmoid-type transition shown by ELM-11. Further analysis for MIL-53(Al) (see Supplementary Note 3) indicated that the LP → NP transition could be represented by a first-order reaction model (Model FO, Eq. (8)), while the NP → LP transition could be represented by a zero-order reaction model (Model ZO, Eq. (9)).

$$\frac{d\alpha}{dt} = k_{FO}(1 - \alpha)\left(P - P_{gate}(\alpha)\right), \qquad (8)$$

$$\frac{d\alpha}{dt} = k_{ZO}\left(P - P_{gate}(\alpha)\right), \qquad (9)$$

where $k_{FO}$ and $k_{ZO}$ are the rate constants for the first- and zero-order reaction models, respectively. Figure 6f, i shows the fitted curves (solid curves), with $k_{FO}$ and $k_{ZO}$ values of 0.07 and 0.015 kPa$^{-1}$ s$^{-1}$, respectively. The gate opening of CuFB was similar to the NP→LP transition of MIL-53(Al) and could therefore be represented by Model ZO (Fig. 6l), with a $k_{ZO}$ of 0.083 kPa$^{-1}$ s$^{-1}$.

An atomistic understanding of these flexible MOFs is also important. Although MIL-53(Al) (wine rack) has a significantly different framework structure than ELM-11 (stacked layer), both exhibit similar transition pathways. MIL-53(Al) exhibits a layer-type structure in terms of the degree of freedom of structural flexibility (indicated by different colours in Fig. 6d)[50–52]. Thus, transitions in each layer proceed stochastically, making the transition rate proportional to the untransformed fraction, $(1 − \alpha)$. However, the pore wall of MIL-53(Al) comprising bdc lacks space for the passage of $CO_2$ through adjacent pores. Therefore, as the reactants do not have an accelerating effect, the LP→NP transition can be represented by Model FO. Here, the stochastic behaviour should lead to a delay in the timing of structural transitions, not only between two layers within a single particle but also between two layers belonging to different particles, which is consistent with the observations obtained from in situ microscopy (see Supplementary Videos 4 and 5). While Model FO for the LP→NP transition is associated with the cooperative deformation of each layer, Model ZO for the gate opening of CuFB can be attributed to the non-cooperative deformation of flexible motifs. In other words, the gate opening of CuFB is induced by the rotation of linkers, causing a localised structural transition from the particle surface. This transition generates a domain boundary between the closed and open phases, which then propagates throughout the interior of a particle at a constant rate that is not dependent on $\alpha$. This behaviour resembles the mass-transfer zone in an adsorption column, leading to the adoption of Model ZO. Although the accuracy of this representation cannot be assessed through in situ microscopy due to the small volumetric change of CuFB during gate opening, microscopic observation indicates that there is no delay in the timing of the structural transition between particles (see Supplementary Videos 6 and 7). This suggests that the transition behaviour of CuFB is not characterised by a stochastic transition (i.e., Model FO). Finally, an interesting finding was that Model ZO could be applied to the NP→LP transition of MIL-53(Al). This suggests that while MIL-53(Al) demonstrates cooperative framework deformation during the LP→NP transition, it exhibits non-cooperative behaviour during the NP→LP transition. Because the NP→LP transition takes place under low-temperature or high-pressure conditions, there are no available insights from in situ microscopy due to equipment limitations. However, one possible explanation for this phenomenon could be the interplay of stabilising and destabilising factors between the two transitions. Specifically, during the LP→NP transition, the framework structure becomes unstable as an offset for stabilisation through guest adsorption. In contrast, during the NP→LP transition, although the strong interactions between the NP structure and guest molecules is lost, the framework structure regains stability as it returns to the LP structure, and the stabilisation through adsorption increases due to an increase in the amount of guest adsorbed. Therefore, for the framework structure, the LP→NP transition is unfavourable, whereas the NP→LP transition is favourable. This discrepancy could impact the domain size in which the framework deforms cooperatively, leading to a change in the reaction model between the two transitions.

## Discussion

This study investigated the structural transition of ELM-11 at varying pressurisation rates by TRXRD and derived a kinetic model through data analysis. The derived model contains a pressure difference term due to $\Delta\Omega^{os}$ and a reaction model term. For the $CO_2$-induced gate opening of ELM-11, the latter term can be represented by the auto-catalytic reaction model. However, this model is not applicable to the two other flexible MOFs analysed (viz. MIL-53(Al) and CuFB) because of differences in the framework structure.

A limitation of this study is that the transition rate was measured only up to 273 K, which is too low as an operating temperature for separation processes. This temperature limit is due to the scale of the pressure gauge used in the current equipment; as the temperature increases, the structural transition occurs at higher pressures. While the rate constants did not significantly change over the range of 223–273 K, the applicability of this trend to the higher temperature

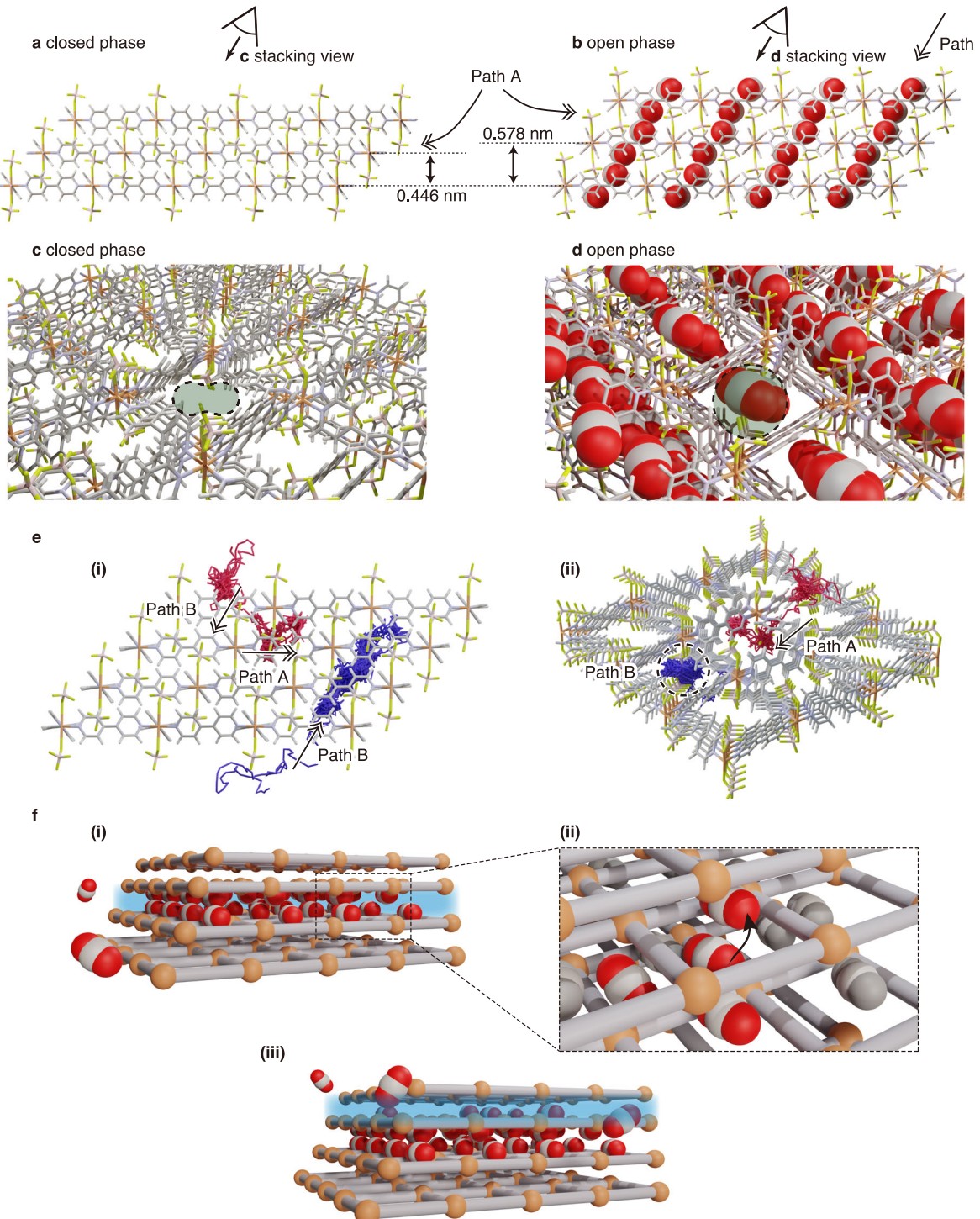

**Fig. 5 | Dynamic mechanism of structural transition based on the atomic structure of ELM-11. a**, **c** Closed- and (**b**, **d**) open-phase atomic structures of ELM-11. **a** and **b** show side views of the closed and open phases, respectively, while **c** and **d** show stacking direction views. **e** Typical trajectories of the centre of mass of a $CO_2$ molecule representing Paths A and B obtained from molecular dynamics simulations. The trajectories are overlaid on the initial structure of ELM-11 in the closed phase (i: side view, ii: stacking direction view). **f** Schematic of the structural transition mechanism of $CO_2$ gate adsorption on ELM-11. Supplementary Video 3 shows the expected movement of $CO_2$. In the crystal structure, the atoms are colour-coded as follows: H (white), B (pink), C (grey), N (purple), O (red), F (green) and Cu (orange).

range typically used for adsorption columns requires further confirmation. Additionally, measuring the desorption rate would be beneficial for enhancing or expanding the developed models. Specifically, Models AC and FO cannot be employed in the desorption process (i.e. 1 to 0 change for $\alpha$) in their present forms, as the term $(1 - \alpha)$ yields 0.

Despite these limitations, this study confirmed that a theoretical framework different from Fick's diffusion equation is required to describe the adsorption kinetics of the guest-induced structural transition of flexible MOFs. The proposed reaction model is closely related to the framework structure flexibility and gas penetration pathways of the MOF. Thus, along with the rate constant, the

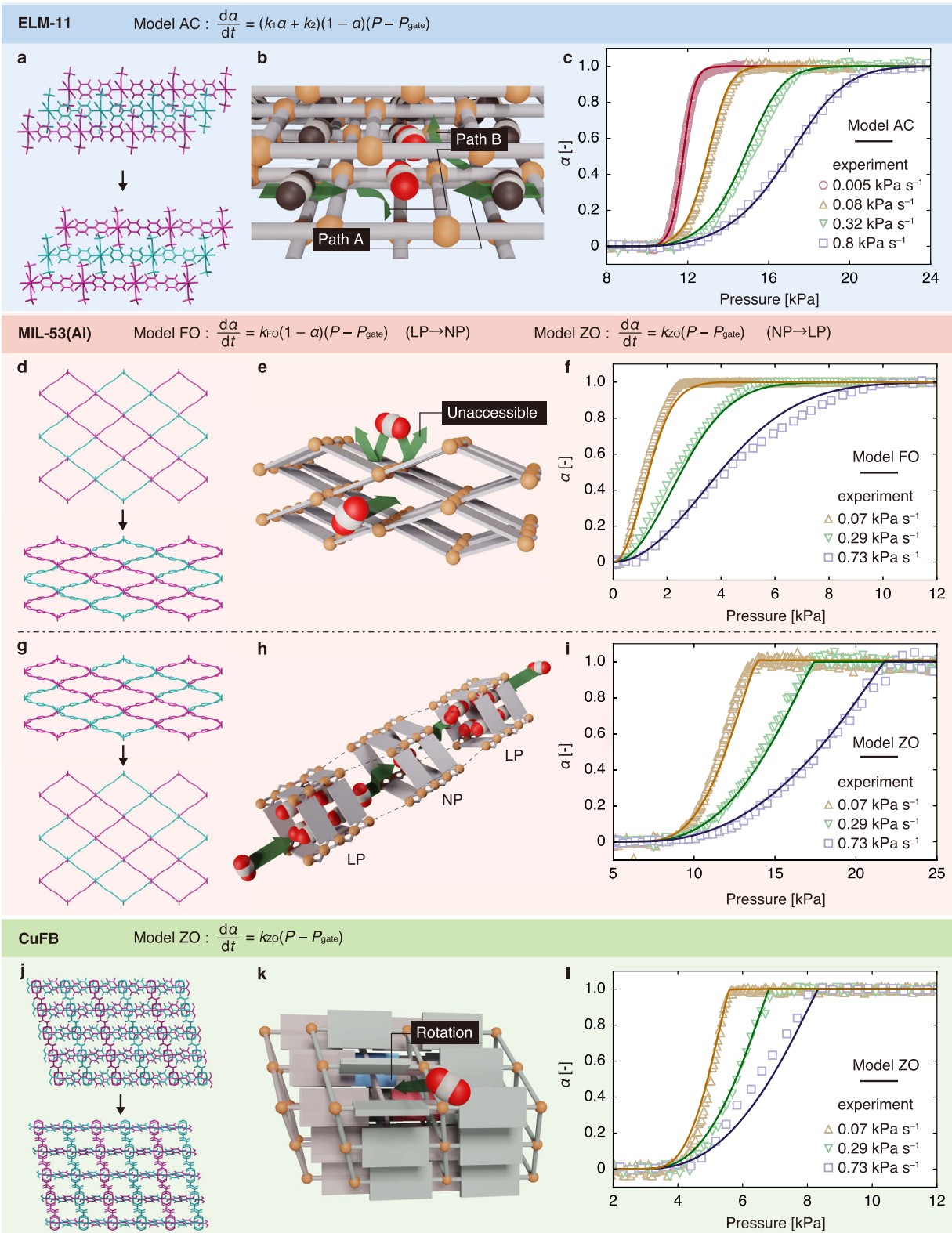

**Fig. 6 | Summary of the structural transition mechanisms of flexible MOFs.**
**a–c**, **d–f**, **g–i** and **j–l** Atomic structure, schematic of the structural transition, and measurement and calculation results for the gate opening of ELM-11 (at 248 K), LP → NP transition of MIL-53(Al) (at 223 K), NP → LP transition of MIL-53(Al) (at 195 K), and gate opening of CuFB (at 248 K), respectively.

structure-dependent functional form of the rate equation influences the adsorption rate of flexible MOFs. For rapid adsorption, the autocatalytic reaction is inherently more advantageous than the first-order reaction, which slows down the progression of structural transition. Therefore, the functional form of the transition rate can be used as a guideline for material design. To facilitate a practical application of this concept, comprehensive studies using a wide range of MOFs must be conducted to systematise the relationship

between the functional form of the transition rate and framework structure. For example, varying the metal ion of MIL-53(Al) is expected to yield equivalent results, while changing the pillar of ELM-11 to a larger one may change the proportions of Path A and Path B by increasing the interlayer distance in the closed state. Furthermore, the functional form of the reaction term may depend on the adsorbed gas molecules: although both Path A and Path B in ELM-11 are available for $CO_2$ molecules, only one pathway may be available for larger molecules. Therefore, systematisation studies must consider the framework structure flexibility, possible penetration paths and target adsorbate. Although the particle size and shape of the MOF are also expected to influence the adsorption rate[53], they are unlikely to affect the structural transition mechanism itself. That is, their contribution can be incorporated into the rate constants, which should be examined through systematic size-controlling synthesis using a microreactor[54]. Combining our method with the direct observation of structural transitions at the atomic level can facilitate the required systematisation with the aid of appropriate experimental and/or computational methods.

## Methods

### Materials

Pre-ELM-11, the hydrated form of ELM-11, was purchased from Tokyo Chemical Industry Co., MIL-53(Al) was provided by SyncMOF Inc., and CuFB was synthesised using a previously reported method[40] (further details can be found in Supplementary Note 4). These samples were observed using scanning electron microscopes (JSM-6700, JEOL and SU8200, Hitachi), and the average particle sizes of pre-ELM-11, MIL-53(Al) and CuFB were found to be $13\,\mu m \times 3\,\mu m$ (acicular shape), $4\,\mu m$, and $4\,\mu m$, respectively (see Supplementary Fig. 8).

### TRXRD measurements

Powder samples were placed at the tip of a 0.5-mm-diameter glass capillary, which was attached to a remote gas handling system with an O-ring. The pre-ELM-11 was evacuated for 12 h at 373 K for transformation into ELM-11. MIL-53(Al) and CuFB were evacuated for 30 min at 573 K and 24 h at 423 K, respectively, for activation. TRXRD patterns were recorded using a flat panel detector at the BL02B2 beamline of the SPring-8 synchrotron facility, Japan. The temperature of the glass capillary was controlled (maintained at 195, 223, 248 and 273 K) using a nitrogen gas blower, and $CO_2$ gas was introduced at a constant flow rate using the gas handling system installed at BL02B2[55]. In situ synchrotron XRD patterns during $CO_2$ gate adsorption were continuously recorded on exposure for 0.5 or 2 s, where the wavelength of the incident X-rays was set to 0.08 nm. After recording 20 XRD patterns of the sample maintained at the initial pressure, $CO_2$ gas was introduced at a specific flow rate (controlled by the mass flow controller) up to a specific pressure.

### Time and pressure evolution of the fraction transformed ($\alpha$)

The time evolution of the recorded XRD patterns, $I_{obs}(2\theta, t)$, was assumed to be a linear combination of the XRD patterns representing the open and closed phases, as follows:

$$I_{calc}(2\theta, t) = \alpha(t)I_{op}(2\theta) + (1 - \alpha(t))I_{cl}(2\theta), \quad (10)$$

where $I_{calc}$ is the intensity of the calculated XRD pattern, and $I_{op}$ and $I_{cl}$ are the intensities obtained by averaging 20 repeated records of the observed values just before and after the measurement, respectively. Moreover, $\alpha(t)$ was determined to minimise the residual sum of squares of the difference between $I_{calc}$ and $I_{obs}$. This linear combination can be applied when the peak intensity and diffraction angle do not change during structural transition, as observed for ELM-11. For MIL-53(Al) and CuFB, changes in both peak intensity and

diffraction angle were observed during structural transition (Supplementary Figs. 5–7). Therefore, a pseudo-Voigt function was used to fit a single peak each time, and $\alpha$ was calculated by normalising its area. Fitting ranges of 4.2–4.7° and 6.6–7.1° were used for MIL-53(Al) and CuFB, respectively.

### In situ microscopy

The volume expansion of flexible MOF particles during $CO_2$ gate adsorption was observed using a digital optical microscope (VW-9000, Keyence Co.). The samples were placed on a cover glass within a vacuum chamber equipped with a glass viewing port. The chamber was connected to a lab-made gas handling system, and $CO_2$ gas was introduced at a constant rate ($0.8\,kPa\,s^{-1}$) at room temperature (approximately 297 K). Movie were recorded at a rate of 30 frames per second. Before measurements, ELM-11, MIL-53(Al) and CuFB were subjected to heat treatment at 373, 473 and 423 K for 12 h, respectively.

### $CO_2$ adsorption measurements and extraction of $h(\alpha)$

The adsorption isotherms of $CO_2$ on ELM-11, MIL-53(Al) and CuFB over the temperature ranges of 223–298, 195–273 and 223–273 K, respectively, were measured using a BELSORP-max gas adsorption analyser (MicrotracBel Co.). Pre-ELM-11 was transformed into ELM-11 by heating it to 373 K for 12 h under vacuum, MIL-53(Al) was activated by heating it to 473 K for 12 h under vacuum, and CuFB was activated by heating it to 423 K for 24 h under vacuum. The adsorbed amount of $CO_2$, $n^*$, was fitted to the theoretical equation for a structural-transition-type adsorption (STA) isotherm[56],

$$n^* = N_{NP}(1 - \sigma) + N_{LP}\sigma, \quad (11)$$

$$\sigma = \frac{y^\beta}{1 + y^\beta}, \quad (12)$$

where $N_{NP}$ and $N_{LP}$ are the equilibrium amounts adsorbed by the NP and LP (open phase) phases, respectively, $\sigma$ is the cumulative log-logistic function, which is an S-shaped function varying around $y = 1$, and $\beta$ is a constant that determines the sharpness of the S-shaped function. The pressure and temperature dependence of $N_i$ ($i = NP, LP$) were modelled using the Sips equation,

$$N_i(P, T) = n_{i0} \frac{(K_{i0}P)^{1/s_{i0}}}{1 + (K_{i0}P)^{1/s_{i0}}}, \quad (13)$$

where $n_{i0}$ is the saturation capacity, $K_{i0}$ is the affinity constant, and $s_{i0}$ is the Sips constant characterising the heterogeneity of the adsorbent surface. The following equations represent the temperature dependence of each parameter:

$$n_{i0} = n_{i,ref} \exp\left[\chi_i\left(1 - \frac{T}{T_{ref}}\right)\right], \quad (14)$$

$$K_{i0} = K_{i,ref} \exp\left[\frac{Q_i}{RT_{ref}}\left(\frac{T_{ref}}{T} - 1\right)\right], \quad (15)$$

$$\frac{1}{s_{i0}} = \frac{1}{s_{i,ref}} + w_i\left(1 - \frac{T_{ref}}{T}\right), \quad (16)$$

where $n_{i,ref}$, $K_{i,ref}$ and $s_{i,ref}$ are the values of $n_{i0}$, $K_{i0}$ and $s_{i0}$ at the reference temperature, $T_{ref}$, $\chi_i$ and $w_i$ are constant parameters, and $Q_i$ is the isosteric heat. When $s_{i,ref} = 1$ and $w_i = 0$, the Sips equation is the same as the Langmuir equation. The pressure and temperature

dependence of $y$ can be represented by the following:

$$\ln y = n_{LP0}s_{LP0}\ln\left[1+(K_{LP0}P)^{\frac{1}{s_{LP0}}}\right] - n_{NP0}s_{NP0}\ln\left[1+(K_{NP0}P)^{\frac{1}{s_{NP0}}}\right] - \frac{\Delta F^{host}}{RT},$$

(17)

where $\Delta F^{host}$ is the Helmholtz free energy change required to deform the host framework from the NP to the LP phase and $R$ is the gas constant. Thus, the following thermodynamic relationship can be written for $\Delta F^{host}$:

$$\Delta F^{host} = \Delta U^{host} - T\Delta S^{host},$$

(18)

where $\Delta U^{host}$ and $\Delta S^{host}$ are differences in the internal energy and entropy of the host framework, respectively, resulting from the deformation of the host. The temperature-dependent STA equation has approximately 15 parameters and is not easy to fit; however, the amount adsorbed (based on the Sips equation) can be separately determined for the NP and LP phases, and the parameters $\Delta U^{host}$, $\Delta S^{host}$ and $\beta$ for the structural transition can be determined later. For ELM-11 and CuFB, the amount adsorbed in the NP phase can be assumed to be 0, reducing the number of parameters to 9. The obtained parameters and resultant theoretical adsorption isotherms are shown in Supplementary Table 1 and Figs. 1, 3 and 4.

Because $\sigma(P, T)$ (Eq. (12)) abruptly changes from 0 to 1 around $P_{gate}$, $h(\alpha, T)$ should be the inverse function of $\sigma(P, T)$.

$$h(\alpha,T) = P_{gate}(\alpha,T) = \sigma(P,T)^{-1}$$

(19)

Notably, because the temperature during TRXRD was controlled using an unstable nitrogen gas blower, the temperature in $\sigma(P, T)$ was fitted as a variable (see Supplementary Note 2).

### Derivation of the pressure difference term from the osmotic free energy ($\Omega^{os}$)

The osmotic free energy $\Omega^{os}$ under isothermal conditions can be represented by the following equation[47]:

$$\Omega^{os}(P) = F^{host} + PV^{host} + \Omega^{guest}(P),$$

(20)

where $P$ is the pressure, $F^{host}$ is the Helmholtz free energy of the host framework, and $V^{host}$ is the volume of the system. Here, $\Omega^{guest}$ is the grand potential of the adsorbed guest, evaluated by integrating a continuous adsorption isotherm, $n^{guest}$, which can be represented as follows:

$$\Omega^{guest}(P) = -\int_0^P n^{guest}(P')V_m(P')dP',$$

(21)

where $V_m$ is the molar volume of the external gas ($V_m = (\partial\mu/\partial P)_T$, where $\mu$ is the chemical potential of the gas).

Considering a first-order Taylor expansion at around $P = P_{gate}$, $\Omega^{os}$ can be approximated by the following:

$$\Omega^{os}(P) \simeq \Omega^{os}(P_{gate}) + \left.\frac{\partial\Omega^{os}}{\partial P}\right|_{P=P_{gate}}(P - P_{gate}),$$

(22)

$$\left.\frac{\partial\Omega^{os}}{\partial P}\right|_{P=P_{gate}} = 0 + V^{host} + \left.\frac{\partial\Omega^{guest}}{\partial P}\right|_{P=P_{gate}},$$

(23)

$$\left.\frac{\partial\Omega^{guest}}{\partial P}\right|_{P=P_{gate}} = n^{guest}(P_{gate})V_m(P_{gate}).$$

(24)

Therefore, the osmotic free energy change between the LP and NP states, $\Delta\Omega^{os}$, is expressed as follows:

$$\Delta\Omega^{os}(P) \simeq \Omega_{LP}^{os}(P) - \Omega_{NP}^{os}(P)$$

$$= \left[\Omega_{LP}^{os}(P_{gate}) + \left.\frac{\partial\Omega_{LP}^{os}}{\partial P}\right|_{P=P_{gate}}(P - P_{gate})\right]$$

$$- \left[\Omega_{NP}^{os}(P_{gate}) + \left.\frac{\partial\Omega_{NP}^{os}}{\partial P}\right|_{P=P_{gate}}(P - P_{gate})\right]$$

$$= \left[\Omega_{LP}^{os}(P_{gate}) - \Omega_{NP}^{os}(P_{gate})\right]$$

$$+ (P - P_{gate})\left\{\left(V_{LP}^{host} - V_{NP}^{host}\right)\right.$$

$$\left. - \left[n_{LP}^{guest}(P_{gate})V_m(P_{gate}) - n_{NP}^{guest}(P_{gate})V_m(P_{gate})\right]\right\}$$

$$= (P - P_{gate})\left\{\Delta V^{host} - V_m(P_{gate})\Delta n^{guest}(P_{gate})\right\},$$

(25)

where $\Delta V^{host}$ and $\Delta n^{guest}$ are the changes in the volume of the host framework and the amount adsorbed, respectively. Here, $P_{gate}$ indicates the pressure at which $\Delta\Omega^{os}$ is zero, i.e. $\Omega_{LP}^{os} = \Omega_{NP}^{os}$.

### Molecular dynamics simulations

To investigate the $CO_2$ diffusion pathways within the ELM-11 framework, molecular dynamics simulations were conducted using the Matlantis software (https://matlantis.com). A cluster model (Supplementary Fig. 9) consisting of four stacked layers, each containing a $3\times3$ grid of bpy, was extracted from the crystallographic structure of closed ELM-11[28]. It was placed at the centre of a $10\times10\times10$ nm simulation box. $CO_2$ molecules were then introduced around the cluster model by the grand canonical Monte Carlo method (details of the method, including the forcefield parameters, are provided in ref. 28), and the pressure and temperature were set to 500 kPa and 273 K, respectively. Using this model as the initial structure, *NVT* simulations were conducted for 0.1 ns, with a timestep of 1 fs at 300 K, using PreFerred Potential[48] version 4.0.0 in the Matlantis software integrated into the Atomic Simulation Environment[57]. After the simulation, the trajectories of the $CO_2$ molecules were analysed, and typical trajectories representing Path A and Path B were overlaid on the initial structure of ELM-11 (Fig. 5e).

### Atomic structures of flexible MOFs

The atomic structure of ELM-11 before and after $CO_2$ adsorption/desorption was previously elucidated by the combined use of Rietveld analysis and molecular simulation[28]. The distances between the component atoms were calculated from the structural data. The atomic structures of MIL-53(Al) in the NP and LP phases were obtained from a previous study[39]. For CuFB, we estimated the closed and $CO_2$-encapsulated structures using molecular simulation-assisted structural analysis based on the structure reported for different guest molecules[40,41]. The details are provided in Supplementary Note 5.

### Data availability

The data that support the findings of this study are provided in Supplementary Information and the online repository at https://github.com/2koza/framework-dependent-kinetics. They can also be obtained from the corresponding author upon request.

### Code availability

The codes used in this study are provided in the online repository at https://github.com/2koza/framework-dependent-kinetics. They can also be obtained from the corresponding author upon request.

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

## Acknowledgements

S.H. thanks Prof. H. Tanaka for fruitful discussions and Dr. A. Hori for providing MIL-53(Al). We are grateful to Mr. Arima and Mr. Saitoh for their assistance in performing the TRXRD measurements. This study was financially supported by a Grant-in-Aid for JSPS Fellows (grant no. 21J14297), Grant-in-Aid for Challenging Research (Pioneering) (grant no. 21K18187), Grant-in-Aid for Scientific Research (B) (grant nos. 22H01848 and 23H03673) and JST CREST (grant no. JPMJCR17I3). The synchrotron radiation experiments were performed at the BL02B2 beamline of SPring-8 with the approval of the Japan Synchrotron Radiation Research Institute (JASRI) (proposal nos. 2020A1666, 2021A1588, 2021B1792, 2022A1755, 2022B1892, 2022B1578, 2022B0555, 2023A1701 and 2023A1696).

## Author contributions

Y.S., S.H., M.T.M. and S.W. designed this study and wrote the paper. Y.S., S.H., I.S., S.K. and H.U. performed the TRXRD measurements. Y.S. and I.S. analysed the experimental data and derived the kinetic model. H.U. synthesised CuFB. S.H. performed the molecular simulations. All authors discussed the results and commented on the manuscript.

## Competing interests

The authors declare no competing interests.
