## [Peer Review File · Nature Communications]

Reviewers' Comments:

Reviewer #1:

Remarks to the Author:

This manuscript presents a study on the temporal structural responses of flexible metal-organic frameworks (MOFs) under varying pressure conditions, specifically focusing on ELM-11, MIL-53, and CuFB. The authors employ time-resolved synchrotron XRD measurement techniques during the pressurization process to investigate the CO₂ sorption kinetics of these flexible MOFs. The results reveal distinct sorption kinetics for each MOF: ELM-11 exhibits autocatalytic-type sorption, MIL-53 follows a first-order model, and CuFB demonstrates a zero-order model. Importantly, this manuscript emphasizes the significance of diffusion pathways in contrast to the conventional bulk diffusion model. Given the scarcity of time-resolved studies on flexible MOFs, this manuscript represents a valuable contribution to the existing literature. Furthermore, the authors conduct a comprehensive analysis of the observed kinetic data through theoretical investigations involving energetics calculations and molecular dynamics simulations, providing an atomic-level interpretation of the findings.

The reviewer has some comments that need clarification.

(1) The accuracy of the fitting model was somewhat ambiguous because in their previous paper published in *Nature Commun.* 11, 3867 (2020), the same author successfully fits the CO₂ sorption kinetics of the same ELM-11 using KJMA equations.

(2) Based on their calculations using molecular dynamics simulations with a small cluster model of a 3 x 3 grid of bpy, the authors proposed the existence of a temporary semi-opening state when ELM-11 undergoes gate-opening behavior towards CO₂. The authors present an atomistic description of the transition state, where one layer is filled with CO₂ and the other is not. While this hypothesis is intuitively understandable, it seems to lack support from other experimental findings. It would be beneficial to provide additional evidence for this mechanism, such as the activation energy of this process and/or the energetics of the semi-opening states.

(3) Do the kinetic rate constants k_1 and k_2 correspond to the rate constants in the diffusion paths A and B? In figure 4, the authors indicated that these rate constants are not affected by temperature. What is the reason for neglecting temperature dependence in these rate constants?

(4) The authors suggested that the CO₂ adsorption kinetic model in MIL-53 differs between adsorption (first-order model) and desorption (zero-order model). The author should provide some discussion on why their kinetics differ.

(5) The kinetics and threshold pressure of the gate-opening behaviors of flexible MOFs are frequently influenced by the sizes of the crystallite particles (ref. *Front. Chem.* 9:772059. doi: 10.3389/fchem.2021.772059). Therefore, it is advisable to include the crystal sizes of ELM-11, MIL-53, and CuFB in this study. Furthermore, it is recommended to discuss the impact of crystal size on the kinetic rate constant in this work.

Reviewer #2:

Remarks to the Author:

The work by Watanabe et al. studies mainly the CO₂ adsorption kinetics of ELM-11, a Cu based metal-organic framework, and also 2 additional typical flexible MOFs.

The analysis of adsorption kinetics of flexible MOFs is a highly important topic and one of the open questions in the field of flexible MOFs. Flexible MOFs are currently discussed as high potential candidates for gas separation, storage and sensing. Manipulating adsorption kinetics is an important target for engineering adsorption separations based on MOFs.

The kinetics of CO₂ adsorption are analyzed here with varying pressurization rates via in situ XRD measurements at the synchrotron with a time resolution of seconds.

For ELM an autocatalytic transformation mechanism is detected and a generalized equation for the kinetics is proposed. However, the authors identify for MIL-53 and CuFB that the model deviates from that of ELM-11.

Overall this is a valuable contribution to the field of flexible MOFs!

Minor comments:

- P6. „TRXRD reflects single particle transition kinetics.“, recently Miura et al. showed that this is

not necessarily the case. In fact, the individual particles (crystals) switch much faster but the crystals have different induction time and induction time distribution (doi.org/10.1002/adma.202207741), the situation may be similar for ELM-11. The ensemble measurement does not allow to make this conclusion stated on P6.

- P7: $h(\alpha)$ etc. are used in equation 4 etc. and discussed but the terms are introduced only later. It would be better to introduce the terms earlier for the reader, before discussing the equations.

- $P_{gate}(\alpha)$: This number is not so well defined, especially when the gating is not sharp. An alternative measure for P_{gate} is the "Adsorption Pressure at half maximum uptake" (APHM).

- „Comparison with MIL53“, end of P11: The NP-OP transition discussion seems not so clear! „propagate outward“? In fact, to open the pore, a lot of molecules have to diffuse from outside to inward to reopen the crystal; in this context diffusion limitation of the narrow and filled, crowded pore probably plays an important role!

Overall this is a valuable and thorough contribution at high scientific level!

Reviewer #3:

Remarks to the Author:

The authors study gate opening phenomenon during CO₂ adsorption on ELM-11 MOF by in situ time-resolved X-ray powder diffraction (TRXRD) and suggest a kinetic model of the transition dynamics, which treats CO₂ propagation as an autocatalytic reaction. The method is also applied to breathing transition in CO₂-MIL-53(AL) and gate opening transition in and CuFB MOF. The experimental data on the dynamics of structural transitions depending on the pressure increase rate is unique and provide a foundation for verification of theoretical and simulation models.

The authors propose an original kinetic model (Eq.6) that is similar to autocatalytic reaction models. The rate of close-the-open phase transformation is proportional the fraction of close cells and to the fraction of open cells, which accelerate the process. This equation implies a linear pressure excess (above the gate opening pressure) dependence. The advantage of this equation is in its simplicity; it contains only 2 fitting parameters and can be easily solved. The authors demonstrate that this equation satisfactory fits the experimental data at 3 different temperatures. I recommend publication after revisions.

Comments.

1. Gate opening and breathing transitions in MOFs exhibit prominent hysteresis between adsorption and desorption isotherms. The authors present only adsorption isotherms. Have the desorption was studied? It would be instructive for authors to present the desorption isotherms for all systems considered.

2. The proposed kinetics models for MIL-53 and CuFB do not imply autocatalysis and are considered as 1st order and 0- order irreversible reactions. The authors refer to the method section for justification, which I could not find. This should be clarified.

3. Breathing transition in MIL-53 was considered in great details in earlier literature. Supplementary Figure 3 shows adsorption isotherms at different temperatures from 195 to 273 K, all of which show the LP-NP transition at low pressure, but the second NP-LP transition is present only for $T < 248$ K. This should be commented: at 248 and 273 the NP=LP transition occurs at $P > 100$ kPa. What was the pressure increase rate in these experiments? Same for all temperatures?

4. Supplementary Figure 3: shows XRPD patterns for LP phase (0 kPa), NP phase (10 kPa), and LP with CO₂ at 100 kPa. Does NP phase at 10 kPa contain CO₂? This should be clarified.

5. The manuscript has to be carefully proofread. There are many inconsistent phrases like "The reaction model of ELM-11 is an autocatalytic reaction with two pathways for CO₂ penetration of

the framework" in the abstract.

Reviewer #1:

This manuscript presents a study on the temporal structural responses of flexible metal-organic frameworks (MOFs) under varying pressure conditions, specifically focusing on ELM-11, MIL-53, and CuFB. The authors employ time-resolved synchrotron XRD measurement techniques during the pressurization process to investigate the CO₂ sorption kinetics of these flexible MOFs. The results reveal distinct sorption kinetics for each MOF: ELM-11 exhibits autocatalytic-type sorption, MIL-53 follows a first-order model, and CuFB demonstrates a zero-order model. Importantly, this manuscript emphasizes the significance of diffusion pathways in contrast to the conventional bulk diffusion model. Given the scarcity of time-resolved studies on flexible MOFs, this manuscript represents a valuable contribution to the existing literature. Furthermore, the authors conduct a comprehensive analysis of the observed kinetic data through theoretical investigations involving energetics calculations and molecular dynamics simulations, providing an atomic-level interpretation of the findings. The reviewer has some comments that need clarification.

We thank Reviewer #1 for their positive remarks.

(1) The accuracy of the fitting model was somewhat ambiguous because in their previous paper published in *Nature Commun.* 11, 3867 (2020), the same author successfully fits the CO₂ sorption kinetics of the same ELM-11 using KJMA equations.

We thank Reviewer #1 for this insightful comment. As previously reported by Finney et al. (*Chem. Mater.* 21, 4692–4705 (2009)), it is known that the KJMA equation yields a curve similar to that of an autocatalytic reaction. This is due to the presence of a term involving the power of time in the KJMA equation, which signifies the acceleration of the reaction as time progresses. This concept aligns with that of an autocatalytic reaction. Therefore, the findings of our current study, which indicates that the transition rate of ELM-11 can be explained by an autocatalytic reaction, is not in conflict with our previous report. To clarify this, the following sentences have been added to the revised manuscript (end of the "**Development of a transition kinetic model**" section):

Note that this result is not contradictory to our previous report, which stated that the transition rate of ELM-11 can be explained by the KJMA equation¹⁴. This is because the autocatalytic reaction and KJMA equation share the same concept: the $k_1\alpha$ term in the autocatalytic reaction and the term involving the power of time (t^n) in the KJMA equation indicate the acceleration of the reaction as time progresses. Therefore, it is known that the KJMA equation yields a curve similar to that of an autocatalytic reaction⁴⁶.

(2) Based on their calculations using molecular dynamics simulations with a small cluster model of a 3 x 3 grid of bpy, the authors proposed the existence of a temporary semi-opening state when ELM-11 undergoes gate-opening behavior towards CO₂. The authors present an atomistic description of the transition state, where one layer is filled with CO₂ and the other is not. While this hypothesis is intuitively understandable, it seems to lack support from other experimental findings. It would be beneficial to provide additional evidence for this mechanism, such as the activation energy of this process and/or the energetics of the semi-opening states.

We thank Reviewer #1 for this insightful comment. In our recently published work (Arima et al., *ACS Appl. Mater. Interfaces* 15, 36975–36987 (2023)), we performed grand canonical molecular dynamics simulations on a simplified layer-stacked MOF particle model and found that the coexistence of closed and open layers within the particle is a thermodynamically favorable state during gate opening. Specifically, we found that the activation energy for the sequential transition mode, where one layer opens to another, is less than 1/45th of the activation energy required for the simultaneous deformation of all layers in the particle. In addition, the in situ microscopy measurements of the current study (refer to Fig. 2 and Supplementary Videos 1 and 2) demonstrated that the volume of the ELM-11 particle

gradually increases over time. We have added the following sentences to the revised manuscript (end of the "Dynamics of structural transition" section):

That is, Model AC suggests that the closed-to-open transition of ELM-11 occurs gradually, layer by layer, through the coexistence of closed and open layers, which is consistent with the experimental observations obtained using in situ microscopy (see Fig. 2 and Supplementary Videos 1 and 2) and a recent theoretical study employing a simplified layer-stacked MOF model⁴⁹.

(3) Do the kinetic rate constants k_1 and k_2 correspond to the rate constants in the diffusion paths A and B? In figure 4, the authors indicated that these rate constants are not affected by temperature. What is the reason for neglecting temperature dependence in these rate constants?

As highlighted by Reviewer #1, k_1 and k_2 can be assumed to correspond to the rate constants in diffusion paths A and B, respectively. We believe that this assumption can be verified by carefully adjusting the aspect ratio of ELM-11 particles. Regarding the negligible temperature dependence of the rate constants, we do not have conclusive evidence at the moment. However, we speculate that $P_{\text{gate}}(\alpha)$ is entirely responsible for the temperature dependence that should be present in equation (6). Specifically, as the temperature increases, not only does the threshold pressure for gate opening increase, but the steepness of the S-shaped curve decreases. The broadening of the S-shaped curve implies an increase in the number of domains that possess greater kinetic energy, allowing them to undergo transition at lower pressures compared to most others. While such an effect is typically considered in rate constants for other kinetic processes, in equation (6), it is encompassed by $P_{\text{gate}}(\alpha)$ rather than k . Consequently, this might lead to an almost negligible temperature dependence in k . We plan to confirm this hypothesis in future research by analyzing the kinetic behavior of a diverse range of flexible MOFs.

(4) The authors suggested that the CO₂ adsorption kinetic model in MIL-53 differs between adsorption (first-order model) and desorption (zero-order model). The author should provide some discussion on why their kinetics differ.

We apologize for any confusion that may have arisen, but it seems that there may have been a misunderstanding. The manuscript describes the first-order model for the LP→NP transition and the zero-order model for the subsequent NP→LP transition. Thus, the NP→LP transition does not refer to the desorption process but rather to the second adsorption process in the breathing phenomenon (the LP→NP→LP transitions, as illustrated in Supplementally Fig. 3). To make this clearer, the transitions are specified at their first occurrences as follows:

MIL-53(Al) exhibits a breathing phenomenon in which the large pore (LP) phase changes to the narrow pore (NP) phase (LP→NP transition) and back to the LP phase (NP→LP transition) along with CO₂ adsorption.

It should be noted that we were unable to measure the desorption process due to equipment limitations, specifically the inability to decrease pressure at a constant rate. However, we acknowledge the importance of studying this process and mention it as a topic for future research in the Discussion section. In addition, we hypothesized that the difference in the transition models between the LP→NP and NP→LP transitions is attributed to the interplay of stabilizing and destabilizing factors between the two transitions, as explained in the manuscript.

(5) The kinetics and threshold pressure of the gate-opening behaviors of flexible MOFs are frequently influenced by the sizes of the crystallite particles (ref. *Front. Chem.* 9:772059. doi: 10.3389/fchem.2021.772059). Therefore, it is advisable to include the crystal sizes of ELM-11, MIL-53, and CuFB in this study. Furthermore, it is recommended to discuss the impact of crystal size on the kinetic rate constant in this work.

We thank Reviewer #1 for this helpful comment. We have included scanning electron microscopy (SEM) images of the MOF samples used in our study in the revised Supplementary Information (Supplementary Fig. 8). Additionally, we have added relevant statements regarding the SEM images and particle sizes to the Method section of the revised manuscript as follows:

These samples were observed using scanning electron microscopes (JSM-6700, JEOL and SU8200, Hitachi), and the average particle sizes of pre-ELM-11, MIL-53(Al), and CuFB were found to be $13 \mu\text{m} \times 3 \mu\text{m}$ (acicular shape), $4 \mu\text{m}$, and $4 \mu\text{m}$, respectively (see Supplementary Fig. 8).

Unfortunately, we cannot provide any data regarding the particle size dependence of the transition rate as controlling the particle size of flexible MOFs is a challenging task with many considerations. However, we acknowledge the significance of investigating the relationship between particle size and transition rate. Therefore, we have included a requirement for future research in the Discussion section, as follows:

Although the particle size and shape of the MOF are also expected to influence the adsorption rate⁵³, they are unlikely to affect the structural transition mechanism itself. That is, their contribution can be incorporated into the rate constants, which should be examined through systematic size-controlling synthesis using a microreactor⁵⁴.

Reviewer #2:

The work by Watanabe et al. studies mainly the CO₂ adsorption kinetics of ELM-11, a Cu based metal-organic framework, and also 2 additional typical flexible MOFs.

The analysis of adsorption kinetics of flexible MOFs is a highly important topic and one of the open questions in the field of flexible MOFs. Flexible MOFs are currently discussed as high potential candidates for gas separation, storage and sensing. Manipulating adsorption kinetics is an important target for engineering adsorption separations based on MOFs.

The kinetics of CO₂ adsorption are analyzed here with varying pressurization rates via in situ XRD measurements at the synchrotron with a time resolution of seconds.

For ELM an autocatalytic transformation mechanism is detected and a generalized equation for the kinetics is proposed. However, the authors identify for MIL-53 and CuFB that the model deviates from that of ELM-11.

Overall this is a valuable contribution to the field of flexible MOFs!

We wholeheartedly thank Reviewer #2 for their positive remarks.

Minor comments:

- P6. „TRXRD reflects single particle transition kinetics.“, recently Miura et al. showed that this is not necessarily the case. In fact, the individual particles (crystals) switch much faster but the crystals have different induction time and induction time distribution (doi.org/10.1002/adma.202207741), the situation may be similar for ELM-11. The ensemble measurement does not allow to make this conclusion stated on P6.

We thank Reviewer #2 for sharing this interesting paper with us. First, we have referred to this paper in the revised manuscript as follows:

Because XRD reflects the average collective behaviour of particles, it is impossible to distinguish between the transition of a single particle or variations in the transition time of each particle. In fact, Miura et al. reported a significant discrepancy between the results obtained by TRXRD and microscopic observations in the case of instantaneous vapour dosing of the DUT-8 series⁴³.

In response to this comment, we repeated in situ microscopy analysis using an improved experimental setup to ensure a more precise and accurate measurements. Further details regarding these measurements

can be found in the revised Methods section. Based on our revised findings, we can confidently conclude that, at least for ELM-11, the microscopic observations align with the TRXRD measurements. The followings are the revised Fig. 2 and the accompanying explanation:

Fig. 2c and Supplementary Videos 1 and 2 show the volume expansion of ELM-11 particles because of gate opening when the CO₂ pressure was increased at 0.8 kPa s⁻¹ and room temperature (~297 K). Volume expansion required ~15 s, which was almost the same as the transition time recorded using TRXRD. More specifically, Fig. 2d shows the time evolution of the number of pixels altered from the initial frame (highlighted as red pixels in Fig. 2c), revealing a sigmoidal curve resembling the TRXRD results (Fig. 2a). This trend remains consistent when the same analysis is applied to the region delineated by white lines in Fig. 2c (represented by the green line in Fig. 2d), indicating that the time evolution indicated by TRXRD reflects single-particle structural transition kinetics. The discrepancy between the current findings and those reported by Miura *et al.*⁴³ could be explained by the gradual increase in pressure during our measurements compared to their instantaneous pressure increase, or by the use of different types of flexible MOFs.

Fig. 2 | CO₂ gate adsorption on ELM-11 by two types of in situ measurements. **a**, Time evolution of α at 248 K and 0.005, 0.08, 0.32, and 0.8 kPa s⁻¹. The origin of the x -axis was set as the time at which the gas pressure was 10 kPa. The time scale was changed and a boundary was set at ~90 s because adsorption at 0.005 kPa s⁻¹ took a very long time. **b**, Pressure dependence of α in **a**. For all pressurisation rates, the structural transition began when the gate opening pressure at 248 K (10 kPa) was exceeded. **c**, Optical microscopy snapshots of ELM-11 particles during CO₂ introduction within a time range of 100–115 s at ~297 K. In each image, the pixels that have changed from the first frame (0 s) are coloured red. The corresponding movies (with and without colouring) are shown in Supplementary Videos 1 and 2. **d**, The time evolution of the normalised number of pixels altered from the initial frame (red pixels in **c**); the yellow line was analysed

using the whole picture, while the green line was analysed using the region delineated by white lines in c. Open circles correspond to the points shown in c.

In addition, we have conducted in situ microscopic measurements for MIL-53(Al) and CuFB (see Supplementary Videos 4–7) and have confirmed that these observations are consistent with the TRXRD results and our atomic understanding of transition mechanisms (please see revised the "**Comparison with other flexible MOFs, MIL-53(Al) and CuFB**" section).

- P7: $h(\alpha)$ etc. are used in equation 4 etc. and discussed but the terms are introduced only later. It would be better to introduce the terms earlier for the reader, before discussing the equations.

The terms $g(\alpha)$ and $h(\alpha)$ were defined in the sentence immediately before equation (4), stating that:

Because the slope and x-intercept are functions of α ($g(\alpha)$ and $h(\alpha)$, respectively), ...

However, we have made a minor modification to further clarify this by stating:

Because the slope and x-intercept are functions of α (hereinafter designated as $g(\alpha)$ and $h(\alpha)$, respectively), ...

- $P_{gate}(\alpha)$: This number is not so well defined, especially when the gating is not sharp. An alternative measure for P_{gate} is the "Adsorption Pressure at half maximum uptake" (APHM).

We thank Reviewer #2 for bringing this to our attention. We have modified the sentence in the "**TRXRD measurements**" section as follows:

Under all conditions, the transition commenced at ~ 10 kPa, which was close to the P_{gate} value at 248 K (the adsorption pressure at half-maximum uptake is 10.3 kPa; see Supplementary Fig. 1).

Please note that after the "**Development of a transition kinetic model**" section, P_{gate} is defined as a function of α , not a single threshold value. In other words, $P_{gate}(\alpha)$ incorporates information about the sharpness of the S-shaped curve (refer to Fig. 3e and the "**CO₂ adsorption measurements and extraction of $h(\alpha)$** " section).

- „Comparison with MIL53“, end of P11: The NP-OP transition discussion seems not so clear! „propagate outward“? In fact, to open the pore, a lot of molecules have to diffuse from outside to inward to reopen the crystal; in this context diffusion limitation of the narrow and filled, crowded pore probably plays an important role!

In response to this comment, we have significantly rewritten the discussion regarding the atomistic understanding of the transitions in MIL-53(Al) and CuFB in the "**Comparison with other flexible MOFs, MIL-53(Al) and CuFB**" section. We believe that the revised discussion provides readers with a better understanding.

Overall this is a valuable and thorough contribution at high scientific level!

Reviewer #3:

The authors study gate opening phenomenon during CO₂ adsorption on ELM-11 MOF by in situ time-resolved X-ray powder diffraction (TRXRD) and suggest a kinetic model of the transition dynamics, which treats CO₂ propagation as an autocatalytic reaction. The method is also applied to breathing transition in CO₂-MIL-53(AL) and gate opening transition in and CuFB MOF.

The experimental data on the dynamics of structural transitions depending on the pressure increase rate is unique and provide a foundation for verification of theoretical and simulation models.

The authors propose an original kinetic model (Eq.6) that is similar to autocatalytic reaction models. The rate of close-the-open phase transformation is proportional the fraction of close cells and to the fraction of open cells, which accelerate the process. This equation implies a linear pressure excess (above the gate opening pressure) dependence. The advantage of this equation is in its simplicity; it contains only 2 fitting parameters and can be easily solved. The authors demonstrate that this equation satisfactory fits the experimental data at 3 different temperatures.

I recommend publication after revisions.

We sincerely thank Reviewer #3 for their positive remarks.

Comments.

1. Gate opening and breathing transitions in MOFs exhibit prominent hysteresis between adsorption and desorption isotherms. The authors present only adsorption isotherms. Have the desorption was studied? It would be instructive for authors to present the desorption isotherms for all systems considered.

We thank Reviewer #3 for this helpful comment. We have added the desorption branches to Supplementary Figs. 1, 3 and 4. The data will be accessible through our data repository.

2. The proposed kinetics models for MIL-53 and CuFB do not imply autocatalysis and are considered as 1st order and 0- order irreversible reactions. The authors refer to the method section for justification, which I could not find. This should be clarified.

We apologize for the incomplete information. We have now provided additional analytical details for MIL-53 and CuFB in the Supplementary Information (Supplementary Note 3). Consequently, we have modified the sentence in the manuscript from "Further analysis for MIL-53(AI) (see Methods) indicated that ..." to "Further analysis for MIL-53(AI) (see Supplementary Note 3) indicated that ...".

3. Breathing transition in MIL-53 was considered in great details in earlier literature. Supplementary Figure 3 shows adsorption isotherms at different temperatures from 195 to 273 K, all of which show the LP-NP transition at low pressure, but the second NP-LP transition is present only for T<248K. This should be commented: at 248 and 273 the NP=LP transition occurs at P>100kPa. What was the pressure increase rate in these experiments? Same for all temperatures?

We thank Reviewer #3 for this insightful comment. We have included the statement "the NP→LP transition at 248 and 273 K occurs above 100 kPa" in the caption of Supplementary Fig. 3. About the question regarding the pressure increase rate, it appears that there may have been a misunderstanding. The data presented are adsorption isotherms obtained using an automated volumetric adsorption measurement apparatus. Each data point is collected after a sufficient duration where the pressure remains nearly constant following gas dosing. As such, we do not have control over the pressure increase rate. However, if the question pertains to the TRXRD measurements for MIL-53(AI), we conducted the measurements at rates of 0.07, 0.29, and 0.73 kPa s⁻¹, as indicated in the legend of Fig. 6f and i.

4. Supplementary Figure 3: shows XRPD patterns for LP phase (0 kPa), NP phase (10 kPa), and LP with CO₂ at 100 kPa. Does NP phase at 10 kPa contain CO₂? This should be clarified.

We apologize for any confusion that may have been caused. The NP phase at 10 kPa contains CO₂ molecules. In Supplementary Figs, 5 and 6, we have corrected the labels to "LP (no guest)", "NP ⊃ CO₂", and "LP ⊃ CO₂".

5. The manuscript has to be carefully proofread. There are many inconsistent phrases like “The reaction model of ELM-11 is an autocatalytic reaction with two pathways for CO₂ penetration of the framework” in the abstract.

We are concerned that we could not comprehend what the reviewer found contradictory in this phrase. The statement that the reaction model of ELM-11 was elucidated to be an autocatalytic reaction model, and that this autocatalytic reaction model results from the presence of two pathways for CO₂ penetration in the ELM-11 framework, is indeed the intended meaning. We have thoroughly examined the revised manuscript and can confirm that there are no inconsistent or contradictory sentences.

Reviewers' Comments:

Reviewer #1:

Remarks to the Author:

The authors have successfully addressed all the concerns raised by the reviewer. Therefore, I recommend accepting the current version of the manuscript, provided that the following issue is addressed.

Reviewer #2:

Remarks to the Author:

All comments are carefully addressed. The work can be published.

Reviewer #3:

Remarks to the Author:

The authors properly addressed my comments and I recommend the revised MS for publication

Reviewer #1:

The authors have successfully addressed all the concerns raised by the reviewer. Therefore, I recommend accepting the current version of the manuscript, provided that the following issue is addressed.

We thank Reviewer #1 for their positive remarks. We are concerned that we could not find the “following issue.”

Reviewer #2:

All comments are carefully addressed. The work can be published.

We thank Reviewer #2 for their positive remarks.

Reviewer #3:

The authors properly addressed my comments and I recommend the revised MS for publication

We thank Reviewer #3 for their positive remarks.